# Analysis and Comparison of Aroma Compounds of Brown Sugar in Guangdong, Guangxi and Yunnan Using GC-O-MS

**DOI:** 10.3390/molecules27185878

**Published:** 2022-09-10

**Authors:** Erbao Chen, Shuna Zhao, Huanlu Song, Yu Zhang, Wanyao Lu

**Affiliations:** 1Beijing Engineering and Technology Research Center of Food Additives, School of Food and Health, Beijing Technology and Business University, Beijing 100048, China; 2Beijing Engineering Laboratory of Geriatric Nutrition & Foods, COFCO Nutrition and Health Research Institute Co., Ltd., Beijing 102209, China; 3COFCO Sugar Co., Ltd., Key Laboratory of Quality & Safety Control for Sugar Crops and Tomato, Ministry of Agriculture of the PRC, Changji 831100, China

**Keywords:** non-centrifugal cane sugar (NCS), GC-O-MS, fingerprint, orthogonal partial least squares discriminant analysis (OPLS-DA)

## Abstract

Guangdong, Guangxi and Yunnan are the three provinces in China that yield the most brown sugar, a brown-red colored solid or powdered sugar product made from sugar cane. In the present study, the differences between odor compounds of brown sugar from Guangdong, Guangxi, and Yunnan provinces in China were compared and analyzed by gas chromatography-olfactometry-mass spectrometry (GC-O-MS). A total of 80 odor compounds, including 5 alcohols, 9 aldehydes, 8 phenols, 21 acids, 14 ketones, 5 esters, 12 pyrazines, and 6 other compounds, were detected. The fingerprint analysis of the brown sugar odor compounds showed 90% similarity, indicating a close relationship among the odor properties of brown sugar in each province. Moreover, the orthogonal partial least squares discriminant analysis (OPLS-DA) was performed to identify the compounds contributing to the volatile classification of the brown sugar from three provinces, which confirmed that OPLS-DA could be a potential tool to distinguish the brown sugar of three origins.

## 1. Introduction

Brown sugar, a traditional sweetener with a distinctive flavor, is mainly made from sugarcane through extraction, clarification, and boiling [1]. It is also called non-centrifugal cane sugar (NCS), which does not separate molasses, so it retains the original flavor and nutrients of sugarcane. Brown sugar is rich in flavonoids and phenols that may act as antioxidants and, therefore, exert benefits on organisms [2,3,4]. Furthermore, it exerts immunomodulatory, cytoprotective, anti-carcinogenic, and anti-cancer properties [5].

A study on the physicochemical properties and storage stability of brown sugar revealed darker color, increased water content and water activity, but decreased glucose and fructose contents due to the Maillard reaction [6]. Similarly, a study on the odor components of brown sugar revealed that acetaldehyde, 2-methylbutyraldehyde, 3-methylbutyraldehyde, 2,6-dimethylpyrazine, nonanal, 2,6-diethylpyrazine, 2,3,5-trimethylpyrazine, furfural, 2,3-dimethylpyrazine, decanal, and 2-acetylpyrrole were the primary components based on their relative concentration [7]. Juliana et al. [8] extracted a total of six odor compounds from brown sugar beverages through simultaneous steam distillation-solvent extraction using a mixture of diethyl ether-pentane (1:1, *w*/*w*) as the solvent. Of the six components, 2-methylpyrazine was the key aroma compound in this beverage. Our previous research has proved that heating of syrup was the primary production step affecting the brown sugar flavor because of the production of a large number of pyrazine compounds [9].

Brown sugar has a green and a strong caramel aroma. Some aroma compounds are inherent in sugarcane, while others are produced by microbial metabolism and Maillard reaction. Sugarcane varieties, growing regions, processing methods, storage conditions and other factors will affect the flavor of brown sugar [10]. The composition and concentration of odor compounds and nutrients in sugarcane from different producing areas are different, which leads to great differences in the flavor composition of brown sugar. However, it is difficult to distinguish the origin of brown sugars only by sensory evaluation. As an intuitive and reproducible method, GC-MS analysis has been effectively applied in origin differentiation studies [11]. Li et al. [12] and Zhao et al. [13] used GC-MS to analyze the volatile odor compounds of ham and rice, respectively, and the results proved that GC-MS played an important role in food odor analysis and origin identification.

Previous studies on brown sugar mostly focused on the identification of key aroma, and there is no study on the flavor differences of brown sugar in different regions. Guangdong, Guangxi and Yunnan are the three major producing areas of brown sugar in China. To the best of our knowledge, the discrimination of brown sugar according to origin has not been reported previously. Therefore, the purpose of this study is to (1) identify the odor compounds of the 18 brown sugar samples from Guangdong, Guangxi, and Yunnan using LLE/GC-O-MS; (2) determine the key odor compounds in brown sugar by calculating OAV; (3) establish the fingerprints of brown sugar from three different origins and (4) find out the compounds that cause the difference using OPLS, so as to provide the basis for selecting brown sugar from different regions when producing foods with different flavor characteristics.

## 2. Results and Discussion

### 2.1. Volatile Aroma Components Analysis

A total of 80 odor compounds, including 5 alcohols, 9 aldehydes, 8 phenols, 21 acids, 14 ketones, 5 esters, 12 pyrazines, and 6 other compounds, were detected in 18 samples from three different regions (Table 1). The brown sugar samples from Guangdong, Guangxi and Yunnan contained 72, 60 and 75 odor compounds, respectively. There are four kinds of alcohols in all three regions, but the types of acid compounds are quite different, with Guangdong and Yunnan containing 20 and 19 acid compounds, respectively, while Guangxi contained only 12 acid compounds. The types of pyrazines, aldehydes, ketones and phenols in the three regions are very close. By comparing the odor compounds in the three regions, it was found that the unique odor compounds of the brown sugar samples in Guangdong were 2-acetyl-5-methylpyrazine, 2-methylbutanoic acid and 3-phenylpropionic acid; the unique odor compound in Guangxi was propylene glycol; and the unique odor compounds in Yunnan were 1,3-dimethoxy-2-hydroxybenzene, 3-hydroxyl-2-methyl-4*H*-pyran-4-one, 3-methyl-1,2-cyclopentanedione, 4-methylpentanoic acid and γ-butyrolactone. These unique odor compounds are expected to be important indicators to distinguish the origin of brown sugar samples.

The average contents of odor compounds in brown sugar samples from the three regions are shown in Figure 1. It can be seen that the highest contents of acid compounds were found in all three regions with 25,595.06, 21,632.44 and 25,187.12 ng/g, followed by phenolic compounds with average contents of 111,69.29, 12,115.37 and 11,744.16 ng/g. In contrast, alcohols, esters, pyrazines and ketones had lower contents.

### 2.2. Analysis of Key Aroma Compounds in Brown Sugar Samples

A total of 46 aroma-active compounds were identified in 18 brown sugar samples by olfactometry, including 4 alcohols, 4 aldehydes, 3 phenols, 15 acids, 11 ketones, 7 pyrazines, and 2 other compounds. According to the odor properties of the aroma active compounds, these compounds can be classified into nine types: sweet/caramel, fruity, green/grassy, sour, sweaty/cheese, nutty, roasted, fatty and potato, which indicated that the aroma profile of brown sugar was the result of the synergistic effect of various odors.

In fact, it is the OAV of the aroma compound, and not its amount, that determines the contribution of the aroma compound. Aroma activity is generally defined as compounds with OAVs greater than 1 [14]. Therefore, the calculation of OAV was carried out for aroma compounds that can be sniffed (Table 2). Among the 18 brown sugar samples, 26 compounds with OAV >1 were considered as the key aroma active compounds of the brown sugar samples in this study and contributed to the overall flavor.

Alcohols: Among the four alcohols that can be sniffed, only furfuryl alcohol had OAV >1 and was only found in Guangxi and Yunnan. The content of furfuryl alcohol in Guangxi and Yunnan was 971.50 and 392.70 ng/g, respectively, and it contributed sweet, toast and caramel aroma to brown sugar. Sugar and amino acids react readily at elevated temperatures to form this compound [15]. The furfuryl alcohol contained in soy sauce has been considered to be one of the main components responsible for its odor, exhibiting a caramel scent, which contributes to the overall flavor of the sample [16].

Aldehydes: Among the aldehydes, there are four aldehydes with OAV >1, namely hexanal, (*E*)-2-nonenal, 3,5-dimethoxy-4-hydroxybenzaldehyde and benzaldehyde. (*E*)-2-nonenal and hexanal are probably oxidation products of polyunsaturated fatty acids [17], with high OAV due to their higher concentration and lower odor threshold, and are key aroma compounds among aldehydes, contributing to the green odor of brown sugar. The average content of benzaldehyde in Guangdong was higher than that in Guangxi and Yunnan, and it may be the degradation product of phenylalanine [14], contributing nutty and caramel aromas to the brown sugar. 3,5-dimethoxy-4-hydroxybenzaldehyde showed close OAV in Guangdong and Yunnan, and was higher than that in Guangxi, contributing sweet and nutty aroma to brown sugars. According to Chen, Song, Li, Chen, Wang, Che, Zhang and Zhao [9], 3,5-dimethoxy-4-hydroxybenzaldehyde is formed during brown sugar production, and the difference in content might be related to the raw materials and processing technology.

Ketones: Four ketones with OAV >1 were found in brown sugar samples, including 3-methyl-1,2-cyclopentanedione, 2-hydroxy-3-methyl-2-cyclopenten-1-one, 2,5-dimethyl-4-hydroxy-3(2*H*)-furanone, and 4-hydroxy-5-methyl-3(2*H*)-furanone. 2,5-Dimethyl-4-hydroxy-3(2*H*)-furanone has the highest OAV and contributes a strong caramel flavor to brown sugar, which is most likely formed by the Maillard reaction through deoxy sugars and is most abundant in strawberries [18,19]. 2-Hydroxy-3-methyl-2-cyclopenten-1-one has a strong caramel aroma and is one of the key odor compounds that contribute to the caramel odor in black tea, soy sauce and molasses [20,21,22]. 3-Methyl-1,2-cyclopentanedione was detected only in Yunnan brown sugar with OAV=14, which contributed sweet and bready aroma to Yunnan brown sugar. 2,5-Dimethyl-4-hydroxy-3(2*H*)-furanone was detected in all the three regions’ samples, but the OAV was greater than 1 only in Guangxi brown sugar, which was caused by its high concentration in Guangxi brown sugar.

Pyrazines: Many products possess a distinctive aroma resulting from pyrazines, which are special Maillard reaction compounds [23,24]. Pyrazine is formed by condensing two α-aminocarbonyl compounds and forming a dihydropyrazine, which oxidizes spontaneously to form the pyrazine [23,25,26]. Among the twelve pyrazines detected in the eighteen samples, there are five kinds of pyrazines with OAV greater than 1, namely 2,3,5-trimethylpyrazine, 2,5-dimethylpyrazine, 2,6-dimethyyl-3-ethylpyrazine, 2,6-dimethyl-3-ethylpyrazine and 2-acetyl-6-methylpyrazine. 2,6-Dimethyl-3-ethylpyrazine exhibited the highest OVA due to its low threshold (OT=0.04 ng/g), contributing a strong roasted potato flavor to brown sugar. 2,5-Dimethylpyrazine and 2,6-dimethylpyrazine were previously reported to be key odor compounds in coffee, exhibiting strong roasted and nutty aroma [27].

Acids: A total of 21 kinds of acid compounds were detected in 18 brown sugars, among which the OAV of 11 kinds of acid compounds was greater than 1. Acetic acid, one of the most abundant compounds in brown sugar, had the highest OAV and contributed sour aroma to the samples. 2-Methylbutanoic acid and 3-methylbutanoic acid exhibited a sour aroma and had been reported to be the key aroma components in Japanese sweet rice wine, which played an important role in the overall flavor of sweet rice wine [28]. Benzoic acid, however, has an unpleasant urine-like odor, which may be caused by phenylalanine under the action of phenylalanine ammonia-lyase in plants [29].

### 2.3. Fingerprint Analysis of Sugar Products from Three Different Regions

A food fingerprint can be defined as molecular markers that indicate a characteristic state or condition of food, thus enabling more accurate product identification [30]. Each sample is regarded as a multidimensional space vector. If two samples are more similar, their space will be closer, and the angle between the two samples’ space vectors will be smaller, which leads the cosine of the angle between the two vectors to move closer to 1. Therefore, the similarity of samples can be expressed by the cosine of the included angle. On the contrary, if the difference between the two samples is greater, the cosine of the included angle becomes smaller. In this study, the samples were determined by GC-O-MS, and the odor-active compounds were selected for fingerprint and similarity evaluation.

It is worth mentioning that the similarity of samples becomes higher when the similarity or the cosine of the angle is above 90%. As depicted in Table 3 and Figure 2, of the six samples in Guangdong, except for Guangdong3, the similarity and cosine of the included angle of the other five samples were above 90%. This indicated that the odor properties of Guangdong3 were quite different than the other five samples, which might have happened due to different processing technology.

The cosine of the included angle of six samples in Guangxi was above 90%, and the similarity of Guangxi3 was just less than 90% (89.80%). This result indicated that the odor properties of these six samples in Guangxi were similar, without much difference

Of the six samples in Yunnan, only Yunnan2 had similarity and cosine of included angle lower than 90%, while the other five samples had similarity and cosine of included angle higher than 90%. This result indicated that the odor attributes of the other five samples were similar, but Yunnan2 had significant differences with them.

### 2.4. Verification of Fingerprint

In order to verify whether the fingerprint method is suitable for the analysis of brown sugar, the verification was carried out. Fingerprint verification includes three parts: stability experiment, precision experiment, and repeatability experiment. Following the sample preparation described in Section 2.4, a brown sugar sample was selected and analyzed by GC-MS after 0, 2, 4, 8, 16, and 24 h. Furthermore, the relative standard deviations (RSD) of the relative retention times (RT) and relative peak areas of the odor-active compounds were calculated. The results showed that the RSD of the relative RT of the odor-active compounds was less than 0.3%, and the RSD of the relative peak areas was less than 5%, indicating that the samples were stable within 24 h and met the requirements of the fingerprint method.

A brown sugar sample was extracted and concentrated with the organic solvent, and then the concentration was injected six times consecutively to calculate the RSD of relative RT and relative peak area of the odor-active compounds. These results showed that the RSD of the relative RT of the odor active compounds was less than 0.5%, and the RSD of the relative peak area was less than 6%, indicating that the precision of the instrument was good and met the requirements of the fingerprint method.

Five brown sugar samples were extracted and analyzed for their odor compounds, followed by the RSD of relative RT and relative peak area of the odor active compounds analysis. The results showed that the RSD of relative RT was less than 0.3%, and the RSD of the relative peak area was less than 7%, indicating that they had good repeatability and met the requirements of the fingerprint method.

### 2.5. Orthogonal Partial Least Squares Discriminant Analysis (OPLS-DA)

The fingerprinting analysis of samples from the three origins of Guangdong, Guangxi, and Yunnan revealed that the majority of samples within each province had similar odor types. In addition, a supervised OPLS-DA multivariate statistical analysis method was used to establish a statistical model in order to distinguish odor compounds between Guangdong and Guangxi, Guangdong and Yunnan, and Guangxi and Yunnan.

By conducting OPLS-DA analysis on the brown sugar, a variable importance of projection diagram (VIP) of the model was obtained. A VIP is a vector that summarizes the contribution of a variable to the explanation of the model. Variables with a VIP >1 are generally considered to contribute to the explanation of the model [31,32]. The samples were assessed as independent variables, and the OPLS-DA model was fitted automatically.

The OPLS-DA and VIP results (Figure 3) indicate that the brown sugars from Guangdong and Guangxi were well separated. The brown sugar from Guangdong and Guangxi showed the greatest degree of separation and low intra-group differences, facilitating an accurate exploration of the differences in composition. VIP diagram elucidated that 4-hydroxybenzaldehyde, 3,5-dimethoxy-4-hydroxybenzaldehyde, n-hexadecanoic acid, butanoic acid, acetic acid, 2-methoxy-4-acetylphenol, 2-acetylpyrrole, pentadecanoic acid, furfuryl alcohol, 4-hydroxyacetophenone, etc., were the main contributors to the distinction between Guangdong and Guangxi samples. These compounds were basically aldehydes, acids, ketones, and phenols. Among these, 3,5-dimethoxy-4-hydroxybenzaldehyde and 4-hydroxybenzaldehyde played an important role in classifying Guangdong and Guangxi. 4-Hydroxybenzaldehyde and 3,5-dimethoxy-4-hydroxybenzaldehyde presented a pleasant nutty and creamy odor. Previously, 4-hydroxybenzaldehyde and 3,5-dimethoxy-4-hydroxybenzaldehyde were identified as the major volatile constituents in brown sugars [33]. Acetic acid is also one of the key compounds that can distinguish brown sugar from two provinces. Acetate is a well-known product of the thermal degradation of saccharides, and it is primarily formed during the early stage of the Maillard reaction, under neutral and alkaline conditions. Acetic acid is formed exclusively by hydrolytic cleavage of β-dicarbonyl in hexose-based systems [34].

As shown in Figure 4, OPLS-DA analysis and VIP results indicate that the brown sugars from Guangdong and Yunnan are distinguishable. The principal compounds contributing to this distinction include n-hexadecanoic acid, acetic acid, dibutylphthalate, 2-acetylpyrrole, 2,5-dimethylpyrazine, and 2-methylpyrazine. Of the compounds with VIP greater than 1, pyrazine compounds appeared, which indicated that pyrazine compounds played a significant role in distinguishing brown sugar between Guangdong and Yunnan. The average content of pyrazines in Guangdong and Yunnan was 2897.28 ng/g and 1441.20 ng/g, respectively, and the pyrazine contents in Guangdong samples were higher than in Yunnan. These compounds could impart a popcorn, nutty, and roasted aroma to brown sugar.

Based on the VIP diagram and OPLS-DA analysis of brown sugar between Guangxi and Yunnan (Figure 5), they were well separated. A number of compounds contributed to the differentiation between the two provinces, including 4-hydroxybenzaldehyde, 3,5-dimethoxy-4-hydroxybenzaldehyde, n-hexadecanoic acid, acetic acid, butanoic acid, and 4-hydroxyacetophenone. Of these volatile compounds, the contribution of 4-hydroxybenzaldehyde was the greatest. The average content of 4-hydroxybenzaldehyde in Guangxi was 2728.55 ng/g, while the samples from Guangxi had no odor compounds. The average contents of 3,5-dimethoxy-4-hydroxybenzaldehyde in Guangxi and Yunnan were 926.34 ng/g and 2967.95 ng/g and the contents in Yunnan were significantly higher than in Guangxi. Perhaps these compounds play an important role in distinguishing the sugars from Guangxi and Yunnan.

## 3. Materials and Methods

### 3.1. Materials

Eighteen brown sugar samples from Guangdong, Guangxi and Yunnan were provided by COFCO. These samples were stored in a refrigerator at −80 °C before analysis.

### 3.2. Standards and Reagents

Ether (purity > 99%), dichloromethane (purity > 99%), anhydrous sodium sulfate, 2-methyl-3-heptanone (purity > 99%) and n-alkane (C_7_-C_30_) were purchased from Sigma-Aldrich (St. Louis, MO, USA), and carrier gas (helium) was purchased from Beijing AP Baif Gases Industry Co., Ltd. (Beijing, China).

### 3.3. Extraction of Odor Compounds from Sugars

The odor compounds in brown sugar were extracted by a liquid–liquid extraction (LLE) method according to Chen et al. [33]. In brief, 50.00 g of brown sugar was placed in a triangular flask, 50 mL of distilled water was added to dissolve the brown sugar, then, 50 mL of ether, 50 mL of dichloromethane and 5 μL of internal standard 2-methyl-3-heptanone (81.6 mg/mL) were added, and the mixture was magnetically stirred at 1000 rpm for 10 min. After centrifugation (Hitachi, Japan) for 30 min at 10,000 rpm, the extract containing the volatile aroma compounds was separated by a funnel. Subsequently, 150.0 g anhydrous sodium sulfate was added to the extract and put into a refrigerator at 4 °C to remove water for 12 h, and filtered with a filter paper. A gentle nitrogen stream was used to concentrate the volume into 100 μL, and the odor compounds were extracted and stored at −80 °C for further analysis.

### 3.4. GC-O-MS

Three well-trained panelists conducted a GC-O analysis of the concentrated distillate. The panelists were recruited from Beijing Technology and Business University’s Molecular Sensory Laboratory. To identify and describe the aroma characteristics of the reference compounds, they smelled several concentrations of reference compounds in model solutions 2 h per day before analysis. The training lasted for one month. For the GC-O analysis, wet gas was delivered to the nose using a blank capillary column to improve the sensitivity of the panelists. The aroma perceptions, intensity, and RT were recorded by the panelists. If two or more panelists detected the aroma, an aroma-active compound was identified [35].

To determine the volatile aroma profile of sugars, an Agilent 7890A gas chromatograph (GC) coupled with an Agilent 5977B mass spectrometer (MS) and a sniffing port (Gerstel, Germany) was used. The aroma extract (1 μL) was injected into a DB-Wax column (60 m × 0.25 mm i.d., film thickness 0.25 μm, Agilent J&W) through splitless mode, and the flow rate of the helium carrier gas was maintained at 1.7 mL/min. The oven temperature was initially programmed at 40 °C, further raised to 100 °C at a rate of 4 °C/min, following a gradual increase up to 200 °C at a rate of 3 °C/min for 5 min, and after achieving an ultimate temperature of 230 °C at a rate of 3 °C/min, it was maintained for 10 min. The interface and ion source were set at 250 °C and 230 °C, respectively, while the electron-impact ionization was set at 70 eV, the acquisition range (*m*/z) at 35–350 amu, and the scan rate at 1.77 scans/s. The transmission line temperature of the olfactory detection port (ODP) was maintained at 235 °C.

### 3.5. Qualitative Analysis

The ionization of a molecule in a vacuum produces a characteristic group of ions of different masses. The plot of relative abundance versus mass of these ions constitutes a mass spectrum. The spectrum can be used to identify the molecule. The unknowns were identified by comparing the fragments with the National Institute of Standards and Technology (NIST) MS Spectral Library (Version 2020), by comparing the odor percepts with the database (http://www.thegoodscentscompany.com) and by calculating the linear retention indices (LRIs) using a homologous series of n-alkanes (C_7_-C_30_). The use of multiple methods can increase the accuracy of qualitative results. Using the internal standard area, the resulting peaks were calibrated, and the aroma compound contents were expressed as nanograms per gram of sample [10].

### 3.6. Odor Activity Value (OAV)

In order to evaluate the contribution of each odorant to the overall aroma of brown sugar, the OAV (ratio of concentration to its odor threshold) was calculated [36]. These threshold values were derived from the literature in water [37].

### 3.7. Statistical Analysis

All experiments in this study were conducted in triplicates, and the data were expressed as mean ± standard deviation. The bar graph was drawn by OriginPro 2022 (OriginLab Corp., Northampton, MA, USA), the OPLS-DA analysis was conducted by SIMCA 14.1 (MKS Instruments, Andover, MA, USA), and the tables were organized by Microsoft Excel 2021 (Microsoft Corp., Redmond, WA, USA).

## 4. Conclusions

In summary, a total of 80 odor compounds, including 5 alcohols, 9 aldehydes, 8 phenols, 21 acids, 14 ketones, 5 esters, 12 pyrazines, and 6 other compounds, were detected in 18 brown sugar samples from three different provinces. The fingerprint analysis showed 90% similarity, indicating a close relationship among the odor components of brown sugars from each province without much difference. Further, the stability, accuracy, and repeatability of the fingerprint method were verified, and speculated that the method could meet the requirements of the fingerprint. In the future, fingerprint might have wider applications due to its characteristic of distinguishing geographical origin and food adulteration. Additionally, the OPLS-DA was employed to identify the tracing of brown sugar and to identify the compounds contributing to brown sugars’ volatile classification. The results demonstrated that 4-hydroxybenzaldehyde, 3,5-dimethoxy-4-hydroxybenzaldehyde, n-hexadecanoic acid, and acetic acid were the essential components in distinguishing the sugars from Guangdong, Guangxi, and Yunnan, validating the efficiency of OPLS-DA.

## Figures and Tables

**Figure 1 molecules-27-05878-f001:**
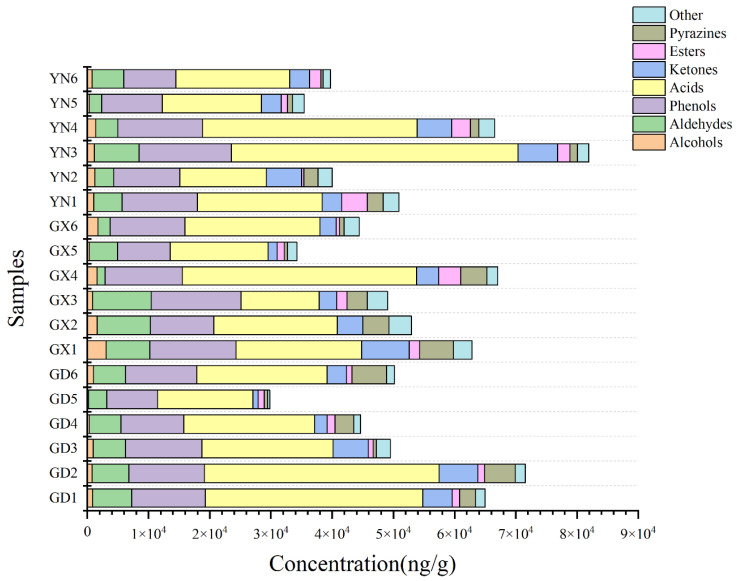
The average content of different kinds of compounds in three regions.

**Figure 2 molecules-27-05878-f002:**
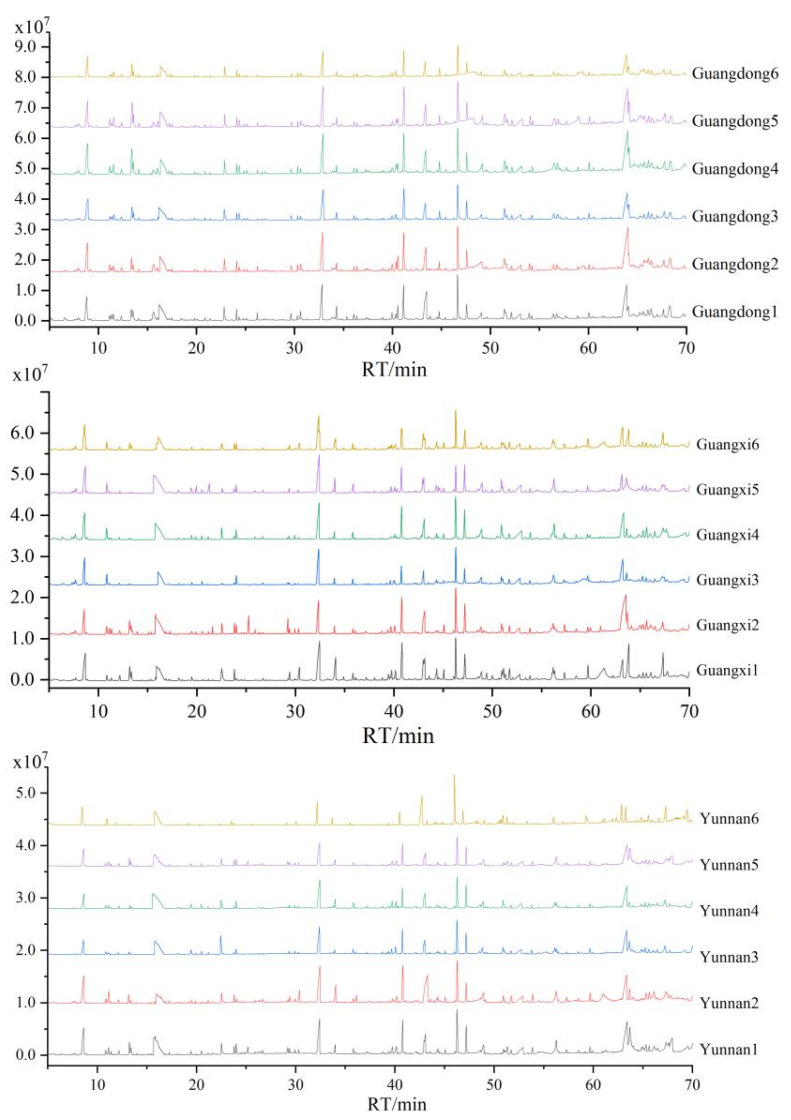
Fingerprint of brown sugar from Guangdong, Guangxi and Yunnan.

**Figure 3 molecules-27-05878-f003:**
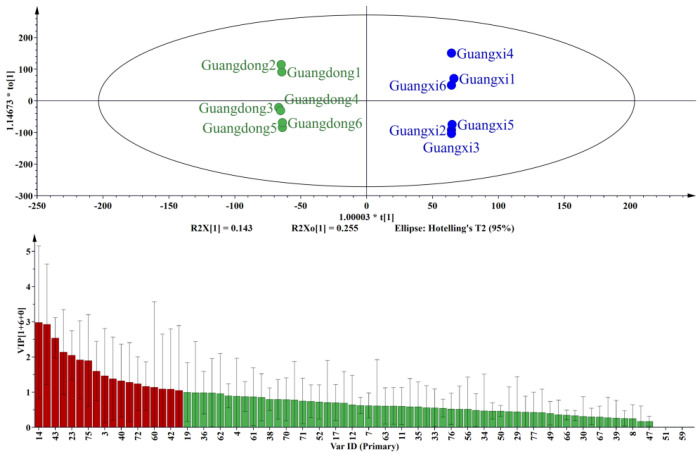
OPLS-DA analysis and VIP diagram of brown sugar in Guangdong and Guangxi.

**Figure 4 molecules-27-05878-f004:**
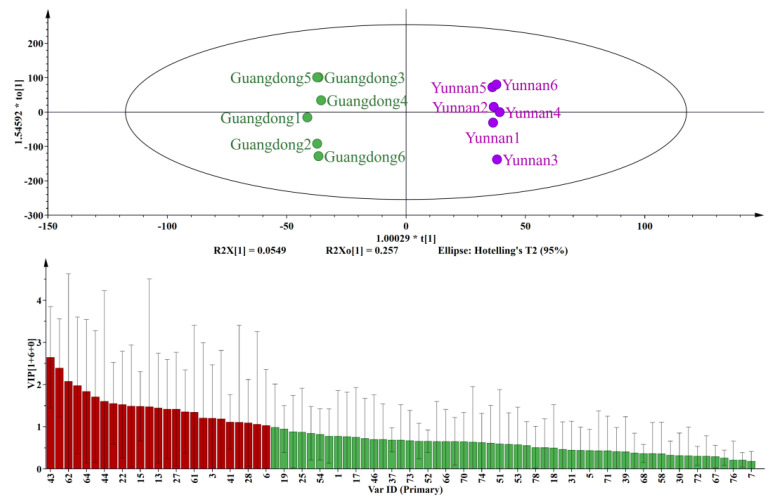
OPLS-DA analysis and VIP diagram of brown sugar in Guangdong and Yunnan.

**Figure 5 molecules-27-05878-f005:**
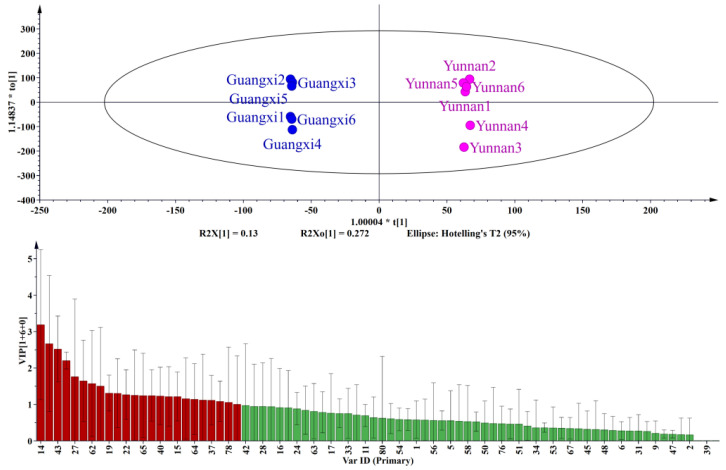
OPLS-DA analysis and VIP diagram of brown sugar in Guangxi and Yunnan.

**Table 1 molecules-27-05878-t001:** Volatile aroma components of brown sugars from different producing areas.

No.	Compounds	Odor description	RI	Identification	Guangdong (ng/g)						Guangxi (ng/g)						Yunnan (ng/g)					
					Guangdong1	Guangdong2	Guangdong3	Guangdong4	Guangdong5	Guangdong6	Guangxi1	Guangxi2	Guangxi3	Guangxi4	Guangxi5	Guangxi6	Yunnan1	Yunnan2	Yunnan3	Yunnan4	Yunnan5	Yunnan6
1	2,3-butanediol	fruit, onion	1568	MS/RI/O	265.09 ± 24.94	184.68 ± 20.31	76.15 ± 9.36	209.60 ± 19.39	83.16 ± 6.00	179.33 ± 13.56	296.97 ± 15.27	-	114.44 ± 10.07	256.59 ± 21.35	81.45 ± 5.45	1019.88 ± 47.28	173.87 ± 12.84	256.31 ± 27.60	466.38 ± 31.83	410.50 ± 35.41	165.52 ± 4.61	50.48 ± 2.32
2	propylene glycol	sweet	1603	MS/RI/O	-	-	-	-	-	-	-	-	75.58 ± 12.71	-	-	-	-	-	-	-	-	-
3	furfuryl alcohol	burnt	1644	MS/RI	474.08 ± 15.80	480.63 ± 13.93	604.00 ± 40.01	130.47 ± 16.31	131.31 ± 6.74	535.66 ± 48.23	2006.99 ± 23.40	1502.75 ± 148.15	675.84 ± 18.94	1027.58 ± 93.40	205.87 ± 11.41	409.98 ± 34.42	881.75 ± 15.61	786.48 ± 36.72	681.50 ± 47.48	589.18 ± 31.74	84.99 ± 5.94	581.65 ± 39.11
4	5-methyl furfuryl alcohol	sweet, caramellic	1705	MS/RI	-	-	-	-	-	295.98 ± 33.01	779.98 ± 180.99	147.60 ± 7.97	-	303.36 ± 121.91	87.16 ± 26.45	325.62 ± 12.36	-	-	-	-	-	189.50 ± 59.19
5	benzyl alcohol	sweet, flower	1865	MS/RI	118.05 ± 16.92	153.98 ± 11.55	299.03 ± 22.63	-	-	17.22 ± 1.07	-	-	-	-	-	-	-	169.70 ± 13.38	-	380.32 ± 22.02	100.30 ± 7.22	-
	Content of total alcohols				857.22	819.29	979.18	340.07	214.47	1028.19	3083.94	1650.35	865.86	1587.53	374.48	1755.48	1055.62	1212.49	1147.88	1380	350.81	821.63
6	hexanal	grass, tallow, fat	1075	MS/RI/O	399.76 ± 21.53	279.98 ± 19.85	313.08 ± 16.32	308.33 ± 6.52	150.05 ± 4.55	260.23 ± 19.05	230.54 ± 19.85	57.51 ± 4.66	130.21 ± 11.55	294.62 ± 15.77	209.07 ± 19.25	182.99 ± 15.58	207.86 ± 7.53	36.23 ± 2.97	231.15 ± 20.79	241.62 ± 17.27	155.67 ± 9.51	182.12 ± 16.88
7	furfural	bread, almond, sweet	1455	MS/RI	-	-	-	50.05 ± 3.69	-	-	97.89 ± 3.70	111.40 ± 5.84	95.82 ± 6.77	-	108.26 ± 5.01	161.29 ± 9.46	-	-	-	-	-	80.62 ± 4.36
8	(*E*)-2-nonenal	cucumber, fat, green	1507	MS/RI/O	56.20 ± 2.12	95.35 ± 2.05	68.21 ± 4.20	71.00 ± 3.80	36.96 ± 3.69	85.51 ± 2.13	-	59.88 ± 0.21	34.18 ± 3.20	60.66 ± 2.67	73.78 ± 5.50	79.64 ± 8.90	44.27 ± 2.08	70.81 ± 6.62	109.32 ± 12.26	80.28 ± 8.59	-	-
9	benzaldehyde	almond, caramel	1514	MS/RI/O	57.98 ± 6.17	109.82 ± 6.24	50.92 ± 3.04	-	-	168.28 ± 3.28	-	-	46.19 ± 5.23	102.28 ± 7.28	52.62 ± 7.13	94.52 ± 2.68	-	49.57 ± 4.48	69.14 ± 6.60	58.67 ± 4.52	54.31 ± 1.43	-
10	5-methylfurfural	almond, caramel	1560	MS/RI	65.10 ± 5.32	60.46 ± 3.14	76.48 ± 2.04	-	-	-	121.80 ± 7.66	58.28 ± 2.64	71.86 ± 6.37	48.57 ± 3.84	27.43 ± 1.07	84.62 ± 4.31	-	74.75 ± 5.59	89.67 ± 8.14	65.65 ± 4.47	-	32.39 ± 2.98
11	2-hydroxymethyl-5-furfural	cardboard	2512	MS/RI	527.94 ± 1.84	543.97 ± 14.99	840.13 ± 26.60	204.68 ± 16.87	-	-	-	835.97 ± 5.84	965.27 ± 12.27	-	387.49 ± 6.35	559.31 ± 6.19	-	182.63 ± 3.23	826.44 ± 9.41	600.61 ± 18.03	-	806.60 ± 45.30
12	4-hydroxy-3-methoxybenzaldehyde	vanilla	2520	MS/RI	1163.20 ± 28.06	938.21 ± 12.55	659.00 ± 12.82	-	930.42 ± 38.08	1330.74 ± 17.04	1123.39 ± 38.79	1146.19 ± 38.02	1256.95 ± 36.57	796.62 ± 26.25	707.06 ± 24.88	813.25 ± 12.95	282.85 ± 23.12	337.09 ± 7.33	1009.19 ± 14.03	694.32 ± 19.89	562.95 ± 26.69	851.26 ± 25.56
13	3,5-dimethoxy-4-hydroxybenzaldehyde	sweet, cocoa, nutty	2905	MS/RI/O	4115.06 ± 183.57	3970.11 ± 230.98	3258.76 ± 103.02	4530.25 ± 115.07	1861.44 ± 137.60	1393.28 ± 152.55	5558.03 ± 199.50	-	-	-	-	-	4106.21 ± 220.69	2356.17 ± 61.68	4986.34 ± 98.27	1882.87 ± 82.12	1255.94 ± 14.65	3220.19 ± 164.10
14	4-hydroxybenzaldehyde	creamy, musty	2908	MS/RI	-	-	-	-	-	1990.86 ± 53.37	-	6374.49 ± 82.32	6982.29 ± 125.16	-	3014.51 ± 138.30	-	-	-	-	-	-	-
	Content of total aldehydes				6385.24	5997.9	5266.58	5164.31	2978.87	5228.9	7131.65	8643.72	9582.77	1302.75	4580.22	1975.62	4641.19	3107.25	7321.25	3624.02	2028.87	5173.18
15	2,6-di-tert-butyl-4-methylphenol	camphor	1904	MS/RI/O	6272.71 ± 42.00	6702.74 ± 109.26	6540.27 ± 49.48	5238.74 ± 73.22	4705.80 ± 38.67	5559.88 ± 55.06	6468.03 ± 79.39	7106.41 ± 74.89	6733.20 ± 53.34	5149.51 ± 72.96	5177.12 ± 46.93	7033.33 ± 25.56	6293.40 ± 47.17	5659.68 ± 67.47	7638.60 ± 69.28	7938.76 ± 88.45	5115.72 ± 52.86	3851.64 ± 64.59
16	4-ethenyl-2-methoxyphenol	clove, curry	2168	MS/RI	3571.11 ± 211.98	3749.85 ± 126.32	3296.40 ± 110.64	3640.61 ± 173.25	2405.55 ± 54.02	3822.00 ± 212.87	3438.47 ± 193.67	1672.14 ± 135.59	4249.96 ± 176.21	4588.34 ± 344.50	1419.61 ± 117.81	2766.58 ± 282.45	3400.23 ± 161.64	3112.76 ± 217.96	4595.74 ± 345.42	3222.71 ± 211.17	2317.34 ± 178.28	1808.79 ± 133.86
17	2,4-di-tert-butylphenol	phenolic	2292	MS/RI	644.13 ± 80.76	456.81 ± 31.92	1052.57 ± 17.01	765.57 ± 39.53	332.60 ± 19.04	947.69 ± 13.41	1292.16 ± 26.55	-	1359.89 ± 78.31	467.47 ± 30.26	323.04 ± 21.63	1207.83 ± 109.14	446.63 ± 20.55	684.35 ± 35.98	675.77 ± 60.09	705.89 ± 49.64	498.28 ± 14.81	558.50 ± 25.05
18	2-methoxy-4-propenylphenol	flower	2250	MS/RI	-	-	-	-	149.00 ± 15.25	-	-	-	-	-	-	-	-	-	-	-	-	391.82 ± 19.80
19	(*E*)-4-propenyl-2-methoxyphenol	flower	2315	MS/RI	630.45 ± 15.19	622.41 ± 11.80	689.57 ± 19.91	-	-	372.06 ± 21.68	605.14 ± 18.49	-	-	-	-	-	619.56 ± 48.92	563.39 ± 38.89	907.11 ± 87.87	662.54 ± 23.60	434.79 ± 35.66	-
20	4-allyl-2,6-dimethoxyphenol	sweet, flower	2510	MS/RI/O	-	-	-	-	-	72.86 ± 9.71	440.73 ± 32.29	-	243.43 ± 15.50	-	-	-	129.33 ± 13.43	-	-	269.60 ± 18.11	132.49 ± 14.66	-
21	4-ethenyl-2,6-dimethoxy-phenol	animal leather	2541	MS/RI	893.16 ± 23.65	799.51 ± 17.03	916.48 ± 36.36	655.80 ± 23.22	321.18 ± 28.46	835.89 ± 38.06	1183.04 ± 31.14	803.61 ± 72.23	1461.19 ± 29.31	1365.67 ± 76.92	746.52 ± 42.58	634.98 ± 13.65	572.43 ± 51.73	807.85 ± 70.45	1249.40 ± 36.45	1069.77 ± 73.55	718.19 ± 34.31	1237.55 ± 79.52
22	2-methoxy-4-acetylphenol	vanilla	2640	MS/RI/O	-	-	-	-	352.34 ± 24.92	-	693.73 ± 55.31	808.51 ± 34.53	638.98 ± 11.43	1094.03 ± 62.69	935.46 ± 64.81	584.08 ± 42.12	855.15 ± 69.12	-	-	-	669.33 ± 59.70	649.86 ± 47.29
	Content of total phenols				12,011.56	12,331.32	12,495.29	10,300.72	8266.47	11,610.38	14,121.3	10,390.67	14,686.65	12,665.02	8601.75	12,226.8	12,316.73	10,828.03	15,066.62	13,869.27	9886.14	8498.16
23	acetic acid	sour	1415	MS/RI/O	10,184.79 ± 464.53	10,757.27 ± 305.67	2778.13 ± 46.39	6350.81 ± 222.21	6432.09 ± 175.47	5900.64 ± 582.42	6650.69 ± 245.63	4210.80 ± 330.79	2977.92 ± 249.84	10,999.92 ± 407.67	5047.46 ± 287.21	11,141.62 ± 393.70	4237.56 ± 172.97	1068.38 ± 53.52	12,748.83 ± 163.89	9209.62 ± 249.19	5570.25 ± 363.18	10,334.03 ± 261.79
24	formic acid	acetic, astringent, fruity	1489	MS/RI/O	249.89 ± 11.00	123.64 ± 7.91	219.04 ± 9.56	195.29 ± 11.26	84.47 ± 6.03	96.96 ± 8.18	-	-	-	-	-	-	91.75 ± 13.52	-	389.25 ± 20.61	426.15 ± 37.81	-	286.39 ± 23.78
25	propanoic acid	pungent, rancid, soy	1526	MS/RI/O	821.24 ± 27.91	619.01 ± 37.09	644.68 ± 16.61	396.79 ± 26.64	303.37 ± 5.15	260.37 ± 16.25	332.73 ± 19.43	147.19 ± 14.68	329.19 ± 2.45	457.22 ± 9.98	280.06 ± 14.46	653.58 ± 28.53	425.26 ± 19.38	299.24 ± 25.79	1143.61 ± 18.50	994.93 ± 49.00	515.40 ± 13.38	344.61 ± 26.73
26	2-methylpropionic acid	rancid, butter, cheese	1563	MS/RI	519.47 ± 21.24	325.82 ± 19.73	264.85 ± 9.33	120.82 ± 9.90	166.03 ± 2.88	4.00 ± 0.24	-	-	88.45 ± 8.92	278.92 ± 25.60	272.22 ± 18.48	322.66 ± 11.67	-	153.86 ± 29.30	621.60 ± 24.32	768.20 ± 37.42	342.05 ± 10.18	-
27	butanoic acid	rancid, cheese, sweat	1607	MS/RI/O	2660.81 ± 89.68	2644.03 ± 91.30	614.21 ± 9.49	2513.84 ± 26.28	1227.91 ± 39.85	1421.39 ± 43.73	810.96 ± 56.86	192.43 ± 16.79	1269.77 ± 80.47	1178.36 ± 29.86	168.86 ± 8.63	440.72 ± 17.86	1266.55 ± 13.63	837.56 ± 53.38	4204.64 ± 71.97	2118.54 ± 37.34	1154.60 ± 33.74	-
28	3-methylbutanoic acid	sweat, acid, rancid	1665	MS/RI/O	1633.63 ± 125.32	1076.75 ± 113.20	645.70 ± 71.56	306.69 ± 15.93	-	12.90 ± 0.34	271.65 ± 16.23	116.04 ± 14.73	218.33 ± 20.09	884.46 ± 15.52	810.07 ± 53.66	710.73 ± 29.15	1123.49 ± 87.06	566.92 ± 16.79	1390.95 ± 38.95	1609.20 ± 36.46	-	-
29	2-methylbutanoic acid	cheese, sweat	1651	MS/RI/O	-	-	-	-	-	373.25 ± 28.50	-	-	-	-	-	-	-	-	-	-	-	-
30	2-methylpentanoic acid	buttery, creamy	1728	MS/RI/O	-	-	-	183.26 ± 5.52	-	37.89 ± 3.86	-	-	-	-	-	-	-	-	-	-	302.40 ± 19.20	-
31	4-methylpentanoic acid	pungent cheese	1820	MS/RI/O	-	-	-	-	-	-	-	-	-	-	-	-	-	-	-	-	119.80 ± 6.83	56.28 ± 4.95
32	hexanoic acid	sweat	1826	MS/RI/O	300.51 ± 7.89	597.73 ± 14.47	487.13 ± 17.43	280.13 ± 27.04	283.56 ± 12.23	435.09 ± 35.20	280.31 ± 18.91	-	155.31 ± 12.61	453.98 ± 23.33	119.62 ± 11.23	112.67 ± 9.39	580.84 ± 13.70	441.59 ± 28.07	727.74 ± 38.34	728.69 ± 15.97	383.25 ± 13.93	78.31 ± 3.80
33	octanoic acid	sweat, cheese	2083	MS/RI/O	228.77 ± 6.47	267.60 ± 21.47	-	-	-	-	-	-	-	-	-	-	-	262.06 ± 13.93	252.12 ± 24.62	487.06 ± 38.82	-	-
34	nonanoic acid	green, fat	2147	MS/RI/O	-	-	-	-	-	413.10 ± 28.63	-	-	-	-	-	-	-	-	-	-	-	227.94 ± 12.15
35	levulinic acid	acidic, sweet, creamy	2312	MS/RI	587.58 ± 22.24	584.31 ± 18.74	589.85 ± 16.89	-	-	-	-	-	306.63 ± 20.02	-	338.16 ± 17.34	530.63 ± 35.66	-	-	653.12 ± 14.73	424.09 ± 35.11	-	-
36	benzoic acid	urine	2392	MS/RI/O	3080.65 ± 130.04	3564.46 ± 241.54	4062.78 ± 303.13	2709.59 ± 234.74	1609.97 ± 79.53	1795.59 ± 129.38	3194.93 ± 194.09	1982.81 ± 136.80	2736.59 ± 235.52	3573.53 ± 145.40	1480.40 ± 134.07	2837.50 ± 175.53	2965.81 ± 197.60	1857.28 ± 192.93	4446.03 ± 192.35	4751.21 ± 272.41	1872.63 ± 65.71	2262.00 ± 138.28
37	dodecanoic acid	metal	2517	MS/RI	555.78 ± 15.49	708.53 ± 18.03	575.78 ± 16.57	215.12 ± 19.98	-	-	-	-	-	-	-	-	-	438.26 ± 33.22	750.95 ± 39.27	504.14 ± 42.56	222.18 ± 21.86	-
38	phenylacetic acid	honey, flower	2551	MS/RI/O	1468.96 ± 56.27	1173.77 ± 75.04	1187.97 ± 92.84	880.89 ± 63.45	730.30 ± 47.81	47.37 ± 3.63	1267.41 ± 64.32	-	945.30 ± 61.58	1180.95 ± 82.76	741.94 ± 40.05	1353.01 ± 89.13	526.12 ± 26.32	903.11 ± 25.21	1487.08 ± 19.13	1976.49 ± 49.24	747.98 ± 54.78	606.57 ± 35.96
39	3-phenylpropionic acid	balsamic	2650	MS/RI/O	-	-	-	156.03 ± 11.33	-	10.79 ± 0.95	-	-	-	-	-	-	-	-	-	-	-	-
40	tetradecanoic acid	sweet, spicy, carnation	2674	MS/RI	821.53 ± 67.39	1273.66 ± 98.64	457.82 ± 19.58	-	-	173.22 ± 16.83	-	-	-	-	-	-	-	498.30 ± 17.36	975.39 ± 21.75	631.33 ± 10.41	252.25 ± 17.50	-
41	pentadecanoic acid	waxy	2784	MS/RI/O	904.37 ± 37.76	1205.19 ± 44.08	867.02 ± 48.07	423.32 ± 20.01	-	806.02 ± 71.41	586.50 ± 17.13	-	-	-	-	-	-	733.11 ± 17.63	1093.45 ± 76.47	886.36 ± 59.46	381.02 ± 26.80	-
42	3-phenyl-2-propenoic acid	balsam, sweet, storax	2815	MS/RI	-	502.70 ± 18.02	-	-	329.63 ± 27.74	555.52 ± 31.40	693.22 ± 19.57	-	-	1366.71 ± 46.48	904.32 ± 31.86	502.43 ± 39.39	-	-	871.20 ± 27.75	491.13 ± 25.65	416.66 ± 18.74	-
43	n-hexadecanoic acid	fatty	2903	MS/RI	11,549.66 ± 283.92	12,914.61 ± 160.72	8039.99 ± 159.00	6586.75 ± 270.87	4432.49 ± 192.77	8965.43 ± 247.84	6422.49 ± 176.26	13,536.74 ± 152.68	3748.63 ± 233.13	17,908.48 ± 240.92	5811.71 ± 96.26	3458.70 ± 197.36	9174.60 ± 92.17	6027.10 ± 231.08	15,086.68 ± 176.07	9036.63 ± 202.89	3926.92 ± 280.51	4354.01 ± 214.68
	Content of total carboxylic acids				35,567.64	38,339.08	21,434.95	21,319.33	15,599.82	21,309.53	20,510.89	20,186.01	12,776.12	38,282.53	15,974.82	22,064.25	20,391.98	14,086.77	46,842.64	35,043.77	16,207.39	18,550.14
44	2-methyl-4,5-dihydro-3(2*H*)-furanone	nutty, creamy	1253	MS/RI/O	341.22 ± 28.42	1177.87 ± 50.20	513.32 ± 23.30	276.67 ± 4.52	70.39 ± 4.46	327.17 ± 28.81	202.04 ± 17.85	291.29 ± 28.07	93.57 ± 6.69	712.73 ± 89.42	141.35 ± 12.32	173.07 ± 3.56	701.59 ± 29.98	1141.42 ± 76.00	391.82 ± 12.71	578.35 ± 18.67	726.04 ± 22.75	1389.22 ± 47.00
45	3-hydroxy-2-butanone	butter, cream	1272	MS/RI	91.84 ± 7.28	126.67 ± 7.48	124.74 ± 9.98	103.62 ± 3.31	-	163.28 ± 2.03	253.98 ± 9.70	145.05 ± 8.21	47.80 ± 1.02	123.45 ± 4.66	32.46 ± 4.69	68.52 ± 1.15	97.36 ± 7.88	74.92 ± 5.14	79.07 ± 3.53	80.41 ± 3.66	56.30 ± 2.47	105.69 ± 8.71
46	1-hydroxy-2-propanone	sweet	1287	MS/RI/O	457.94 ± 39.12	582.30 ± 18.77	605.26 ± 9.93	438.74 ± 12.52	206.31 ± 12.44	652.59 ± 47.59	841.13 ± 59.74	715.56 ± 23.75	503.38 ± 2.41	505.10 ± 33.50	220.33 ± 5.58	211.55 ± 20.44	458.82 ± 7.71	360.04 ± 29.17	382.45 ± 17.01	461.70 ± 11.83	355.89 ± 14.98	438.74 ± 30.24
47	1-hydroxy-2-butanone	oily, alcoholic	1375	MS/RI/O	39.22 ± 2.39	111.06 ± 5.97	41.37 ± 3.74	43.91 ± 3.57	-	16.70 ± 0.68	-	76.40 ± 7.75	82.88 ± 4.40	62.58 ± 4.71	-	24.26 ± 1.31	56.34 ± 6.77	46.29 ± 6.89	63.27 ± 6.83	56.43 ± 3.28	-	-
48	1-acetoxy-2-propanone	fruity, nutty	1451	MS/RI/O	118.22 ± 4.77	230.25 ± 15.43	187.71 ± 12.77	168.77 ± 15.67	-	66.52 ± 6.06	218.30 ± 12.12	159.99 ± 17.64	115.45 ± 11.12	245.41 ± 19.90	139.08 ± 6.91	235.11 ± 24.82	176.14 ± 11.14	191.49 ± 11.15	132.61 ± 7.14	177.62 ± 9.60	125.30 ± 4.14	131.45 ± 4.25
49	4,5-dihydro-5-methyl-2(3*H*)-furanone	sweet, cocoa, woody	1590	MS/RI	65.34 ± 7.38	113.07 ± 5.93	89.27 ± 2.10	-	-	-	-	-	-	-	-	-	-	-	-	90.38 ± 5.27	-	-
50	2(5*H*)-furanone	buttery	1727	MS/RI/O	475.08 ± 10.84	581.86 ± 9.05	479.55 ± 13.75	-	-	145.49 ± 10.40	394.39 ± 11.56	265.72 ± 18.69	286.88 ± 27.60	221.84 ± 19.51	203.23 ± 15.77	363.54 ± 28.70	-	232.91 ± 14.77	712.75 ± 37.46	266.08 ± 24.57	-	154.22 ± 8.91
51	3-methyl-1,2-cyclopentanedione	sweet, maple, bready	1781	MS/RI/O	-	-	-	-	-	-	-	-	-	-	-	-	-	-	-	-	-	361.67 ± 24.53
52	2-hydroxy-3-methyl-2-cyclopenten-1-one	caramellic	1807	MS/RI/O	468.60 ± 24.21	482.48 ± 34.60	707.42 ± 68.00	283.22 ± 26.02	182.61 ± 11.36	588.56 ± 57.85	879.99 ± 59.16	726.67 ± 69.62	745.18 ± 66.72	519.02 ± 24.12	186.46 ± 22.90	476.39 ± 46.95	681.71 ± 49.56	637.34 ± 38.58	716.33 ± 14.99	684.63 ± 28.94	248.07 ± 2.51	-
53	3-hydroxyl-2-methyl-4H-pyran-4-one	caramel	1931	MS/RI	-	-	-	-	-	-	-	-	-	-	-	-	-	-	-	-	201.59 ± 23.74	94.06 ± 7.11
54	2(3*H*)-furanone	cotton candy	2002	MS/RI/O	974.60 ± 62.46	721.47 ± 64.38	1069.93 ± 92.52	351.84 ± 39.07	292.08 ± 13.56	470.52 ± 30.39	631.95 ± 22.11	615.99 ± 36.85	655.84 ± 55.61	705.84 ± 16.73	496.09 ± 31.24	977.50 ± 56.20	560.98 ± 8.63	596.89 ± 59.04	1123.58 ± 46.28	1324.65 ± 17.01	397.11 ± 16.87	323.38 ± 29.23
55	2,5-dimethyl-4-hydroxy-3(2*H*)-furanone	caramel	2012	MS/RI/O	271.76 ± 26.35	326.77 ± 19.05	454.38 ± 18.37	230.03 ± 13.34	123.31 ± 4.96	731.60 ± 15.54	2168.59 ± 55.50	1158.64 ± 13.07	298.58 ± 23.33	471.80 ± 39.52	69.64 ± 3.26	144.05 ± 13.78	415.29 ± 14.57	861.94 ± 90.58	625.51 ± 26.54	529.53 ± 18.15	-	141.69 ± 16.00
56	4-hydroxy-5-methyl-3-(2*H*)-furanone	caramel	2113	MS/RI/O	-	-	-	181.54 ± 21.75	-	19.69 ± 1.35	612.25 ± 48.80	-	-	-	-	-	-	-	-	-	-	99.22 ± 7.93
57	4-hydroxyacetophenone	sweet	2958	MS/RI/O	1461.87 ± 17.59	1892.62 ± 25.11	1469.22 ± 34.48	-	-	-	1546.24 ± 57.40	-	-	-	-	-	-	1656.02 ± 57.98	2234.53 ± 34.42	1370.88 ± 77.57	1132.23 ± 64.94	-
	Content of total ketones				4765.69	6346.42	5742.17	2078.34	874.7	3182.12	7748.86	4155.31	2829.56	3567.77	1488.64	2673.99	3148.23	5799.26	6461.92	5620.66	3242.53	3239.34
58	dimethyl butanedioate	sweet, fruity, green	1558	MS/RI	-	-	-	-	-	-	-	-	-	590.16 ± 16.86	-	-	120.03 ± 12.32	-	-	-	-	-
59	γ-butyrolactone	caramel, sweet	1647	MS/RI	-	-	-	-	-	-	-	-	-	-	-	-	-	-	-	-	-	193.04 ± 34.38
60	dimethyl glutarate	floral	1687	MS/RI	-	-	-	-	-	-	-	-	-	2030.19 ± 99.86	-	-	1039.76 ± 40.69	-	-	-	-	-
61	benzyl benzoate	balsamic, oil, herb	2592	MS/RI	-	-	-	205.81 ± 17.70	-	503.82 ± 28.75	621.78 ± 52.33	-	503.64 ± 41.22	-	431.15 ± 37.87	482.89 ± 31.13	510.18 ± 36.70	381.49 ± 27.23	682.59 ± 34.84	514.18 ± 27.27	-	-
62	dibutyl phthalate	faint odor	2705	MS/RI	1264.25 ± 33.84	1113.96 ± 65.74	810.31 ± 68.93	1061.36 ± 22.53	979.13 ± 65.98	373.31 ± 25.22	1089.36 ± 16.24	-	1202.60 ± 47.72	977.92 ± 29.04	735.97 ± 19.95	-	2530.75 ± 52.67	-	1333.62 ± 38.03	2536.41 ± 162.39	955.08 ± 69.59	1694.09 ± 93.37
	Content of total esters				1264.25	1113.96	810.31	1267.17	979.13	877.13	1711.14	0	1706.24	3598.27	1167.12	482.89	4200.72	381.49	2016.21	3050.59	955.08	1887.13
63	2-methylpyrazine	popcorn	1259	MS/RI	366.92 ± 30.48	662.33 ± 40.09	159.11 ± 11.85	402.41 ± 7.65	112.41 ± 5.11	854.10 ± 77.77	651.18 ± 27.24	500.27 ± 17.30	235.82 ± 8.38	561.03 ± 42.71	76.46 ± 1.56	46.44 ± 5.46	289.74 ± 9.95	196.65 ± 14.01	172.90 ± 15.26	203.79 ± 21.19	88.40 ± 1.38	-
64	2,5-dimethylpyrazine	cocoa, nutty, roast beef	1321	MS/RI/O	924.49 ± 34.20	1938.05 ± 101.57	111.73 ± 10.50	963.30 ± 10.96	201.81 ± 14.82	1827.76 ± 35.72	1343.97 ± 86.27	989.09 ± 53.50	1309.65 ± 15.59	1748.65 ± 121.67	88.92 ± 6.07	70.87 ± 2.23	1114.25 ± 44.64	703.42 ± 34.33	419.79 ± 28.03	430.64 ± 21.63	375.15 ± 3.73	93.56 ± 8.17
65	2,6-dimethylpyrazine	nutty, cocoa, roast beef	1326	MS/RI/O	682.76 ± 22.04	1035.34 ± 80.61	144.76 ± 15.74	983.96 ± 18.81	72.49 ± 6.85	1094.56 ± 26.51	1444.15 ± 48.60	1045.87 ± 46.22	800.00 ± 42.14	946.87 ± 86.05	68.92 ± 14.85	85.72 ± 11.44	693.95 ± 41.35	273.74 ± 19.98	230.77 ± 22.95	289.35 ± 16.10	139.77 ± 2.60	208.04 ± 17.25
66	2,3-dimethylpyrazine	nutty, cocoa, meat	1343	MS/RI	125.46 ± 9.27	231.83 ± 13.14	56.13 ± 2.20	-	38.84 ± 3.35	325.66 ± 36.06	253.09 ± 14.24	142.65 ± 9.66	72.47 ± 5.15	216.09 ± 18.81	10.89 ± 1.38	-	152.68 ± 11.65	73.56 ± 4.48	75.96 ± 6.30	70.09 ± 5.01	42.31 ± 1.02	-
67	2-ethyl-6-methylpyrazine	roasted hazelnut	1382	MS/RI	69.27 ± 2.41	157.44 ± 5.95	-	92.61 ± 1.87	13.81 ± 2.78	88.18 ± 9.04	165.64 ± 5.86	114.68 ± 7.43	58.06 ± 6.05	167.14 ± 12.11	14.91 ± 1.33	19.91 ± 1.53	153.89 ± 9.44	66.03 ± 4.67	52.17 ± 4.62	46.08 ± 2.79	21.31 ± 0.76	5.15 ± 0.38
68	2-ethyl-5-methylpyrazine	fruit, sweet	1376	MS/RI	55.63 ± 6.94	227.21 ± 18.30	-	-	-	-	151.24 ± 10.05	139.24 ± 7.77	123.72 ± 3.04	178.42 ± 6.03	-	21.87 ± 0.65	88.57 ± 8.00	105.86 ± 13.64	129.60 ± 16.42	77.52 ± 8.13	-	-
69	2,3,5-trimethylpyrazine	roast, potato, must	1405	MS/RI/O	-	-	-	180.76 ± 15.90	35.78 ± 3.42	306.68 ± 21.34	249.04 ± 14.64	250.84 ± 14.85	40.86 ± 3.21	-	46.57 ± 3.30	-	-	125.18 ± 7.11	-	-	-	33.86 ± 3.99
70	2,5-dimethyl-3-ethylpyrazine	potato, roast	1445	MS/RI/O	151.21 ± 12.91	315.82 ± 12.23	-	157.83 ± 2.51	-	338.49 ± 32.43	-	-	-	-	-	-	109.14 ± 7.37	201.75 ± 16.35	138.27 ± 14.70	119.00 ± 1.38	-	-
71	2,6-dimethyl-3-ethylpyrazine	potato	1455	MS/RI/O	-	-	-	-	79.78 ± 4.61	186.16 ± 12.73	412.18 ± 36.51	272.34 ± 16.48	203.53 ± 6.05	286.15 ± 18.85	-	-	-	-	-	-	82.88 ± 8.05	-
72	2-methyl-6-vinylpyrazine	hazelnut	1487	MS/RI	-	-	-	-	-	177.67 ± 24.80	688.17 ± 39.07	665.08 ± 41.61	311.55 ± 14.08	-	181.63 ± 8.49	550.08 ± 34.15	-	133.21 ± 8.77	-	-	-	-
73	2-acetyl-5-methylpyrazine	popcorn	1664	MS/RI/O	-	-	-	156.34 ± 13.27	-	218.85 ± 16.63	-	-	-	-	-	-	-	-	-	-	-	-
74	2-acetyl-6-methylpyrazine	coffee, cocoa, popcorn	1673	MS/RI/O	251.42 ± 21.79	419.45 ± 28.68	-	141.33 ± 9.04	-	245.73 ± 14.16	183.55 ± 23.20	144.23 ± 7.83	121.93 ± 11.22	218.58 ± 16.46	-	-	-	408.61 ± 9.22	-	113.67 ± 8.19	96.94 ± 7.01	-
	Content of total pyrazines				2627.16	4987.47	471.73	3078.54	554.92	5663.84	5542.21	4264.29	3277.59	4322.93	488.3	794.89	2602.22	2288.01	1219.46	1350.14	846.76	340.61
75	2-acetylpyrrole	nutty, walnut, bread	1947	MS/RI	679.69 ± 54.14	856.53 ± 39.09	1676.37 ± 66.60	531.22 ± 38.42	111.84 ± 5.12	791.31 ± 37.88	2227.41 ± 88.13	2964.65 ± 58.72	2703.33 ± 101.11	747.17 ± 33.20	461.32 ± 47.83	1263.31 ± 41.84	855.32 ± 80.10	1610.00 ± 150.67	942.57 ± 99.06	1761.17 ± 84.12	1298.42 ± 59.86	971.33 ± 70.56
76	2-acetylfuran	balsamic	1490	MS/RI/O	-	-	-	-	-	5.27 ± 0.85	-	231.24 ± 17.91	103.74 ± 7.62	-	37.99 ± 2.05	72.73 ± 1.11	-	-	-	-	-	54.65 ± 3.48
77	(+)-limonene	citrus, mint	1201	MS/RI	-	-	-	-	48.67 ± 3.12	-	210.31 ± 21.01	-	-	-	71.37 ± 2.85	99.25 ± 2.63	69.37 ± 7.74	69.29 ± 5.81	89.42 ± 6.55	51.53 ± 3.80	-	-
78	phenylethylene	balsamic, gasoline	1247	MS/RI/O	651.29 ± 54.83	613.81 ± 14.74	573.68 ± 15.42	451.99 ± 11.84	160.07 ± 12.09	457.61 ± 37.97	546.32 ± 24.83	469.44 ± 28.01	511.72 ± 22.34	984.77 ± 64.37	984.42 ± 73.21	1025.24 ± 14.21	500.79 ± 9.91	459.13 ± 34.72	564.51 ± 15.57	601.32 ± 18.81	471.01 ± 19.63	162.20 ± 23.00
79	methyl sulfoxide	garlic	1576	MS/RI	151.90 ± 19.28	176.80 ± 8.92	63.28 ± 7.19	89.59 ± 3.82	49.16 ± 4.14	2.89 ± 1.67	-	-	-	-	-	-	-	178.52 ± 18.72	237.91 ± 15.70	207.83 ± 13.45	151.60 ± 18.19	42.30 ± 4.61
80	1,3-dimethoxy-2-hydroxybenzene	medicine, phenol, smoke	2296	MS/RI	-	-	-	-	-	-	-	-	-	-	-	-	1138.32 ± 54.20	-	-	-	-	-
	Content of other compounds				1482.88	1647.14	2313.33	1072.8	369.74	1257.08	2984.04	3665.33	3318.79	1731.94	1555.1	2460.53	2563.8	2316.94	1834.41	2621.85	1921.03	1230.48
	Total identified/detected				64,961.64	71,582.58	49,513.54	44,621.28	29,838.12	50,157.17	62,834.03	52,955.68	49,043.58	67,058.74	34,230.43	44,434.45	50,920.49	40,020.24	81,910.39	66,560.3	35,438.61	39,740.67

**Table 2 molecules-27-05878-t002:** OAV of key odor compounds in brown sugar.

No.	Compounds ^a^	OT (ng/g) ^b^	GD1	GD2	GD3	GD4	GD5	GD6	GX1	GX2	GX3	GX4	GX5	GX6	YN1	YN2	YN3	YN4	YN5	YN6
1	pentadecanoic acid	500	2	2	2	1	-	2	1	-	-	-	-	-	-	1	2	2	1	-
2	2-methylbutanoic acid	20	-	-	-	-	-	19	-	-	-	-	-	-	-	-	-	-	-	-
3	3-methylbutanoic acid	1.8	908	598	359	170	-	7	151	64	121	491	450	395	624	315	773	894	-	-
4	4-methylpentanoic acid	1.9	-	-	-	-	-	-	-	-	-	-	-	-	-	-	-	-	63	30
5	acetic acid	13	783	827	214	489	495	454	512	324	229	846	388	857	326	82	981	708	428	795
6	benzoic acid	1000	3	4	4	3	2	2	3	2	3	4	1	3	3	2	4	5	2	2
7	butanoic acid	20	133	132	31	126	61	71	41	10	63	59	8	22	63	42	210	106	58	-
8	hexanoic acid	4.8	63	125	101	58	59	91	58	-	32	95	25	23	121	92	152	152	80	16
9	nonanoic acid	1.6	-	-	-	-	-	258	-	-	-	-	-	-	-	-	-	-	-	142
10	octanoic acid	22	10	12	-	-	-	-	-	-	-	-	-	-	-	12	11	22	-	-
11	phenylacetic acid	17	86	69	70	52	43	3	75	-	56	69	44	80	31	53	87	116	44	36
12	hexanal	1.4	286	200	224	220	107	186	165	41	93	210	149	131	148	26	165	173	111	130
13	(*E*)-2-nonenal	0.19	296	502	359	374	195	450	-	315	180	319	388	419	233	373	575	423	-	-
14	3,5-dimethoxy-4-hydroxybenzaldehyde	1900	2	2	2	2	1	1	3	-	-	-	-	-	2	1	3	1	1	2
15	benzaldehyde	60	1	2	1	-	-	3	-	-	1	2	1	2	-	1	1	1	1	-
16	3-methyl-1,2-cyclopentanedione	26	-	-	-	-	-	-	-	-	-	-	-	-	-	-	-	-	-	14
17	2-hydroxy-3-methyl-2-cyclopenten-1-one	10	47	48	71	28	18	59	88	73	75	52	19	48	68	64	72	68	25	-
18	2,5-dimethyl-4-hydroxy-3(2*H*)-furanone	1.6	170	204	284	144	77	457	1355	724	187	295	44	90	260	539	391	331	-	89
19	4-hydroxy-5-methyl-3-(2*H*)-furanone	500	-	-	-	<1	-	<1	1	-	-	-	-	-	-	-	-	-	-	<1
20	phenylethylene	37	18	17	16	12	4	12	15	13	14	27	27	28	14	12	15	16	13	4
21	furfuryl alcohol	1415	<1	<1	<1	<1	<1	<1	1	1	<1	1	<1	<1	1	1	<1	<1	<1	<1
22	2,3,5-trimethylpyrazine	23	-	-	-	8	2	13	11	11	2	-	2	-	-	5	-	-	-	1
23	2,5-dimethylpyrazine	80	12	24	1	12	3	23	17	12	16	22	1	1	14	9	5	5	5	1
24	2,6-dimethyl-3-ethylpyrazine	0.04	-	-	-	-	1995	4654	10,305	6809	5088	7154	-	-	-	-	-	-	2072	-
25	2,6-dimethylpyrazine	250	3	4	1	4	<1	4	6	4	3	4	<1	<1	3	1	1	1	1	1
26	2-acetyl-6-methylpyrazine	300	1	1	-	<1	-	1	1	<1	<1	1	-	-	-	1	-	<1	<1	-

^a^ Volatile compounds that can be smelled at sniffer port. ^b^ Odor thresholds were referenced in a book, named: *odor thresholds compilations of odor threshold values in air, water and other media*.

**Table 3 molecules-27-05878-t003:** Fingerprint results of brown sugar from each producing area.

No.	Compounds	Guangdong						Guangxi						Yunnan					
		Guangdong1	Guangdong2	Guangdong3	Guangdong4	Guangdong5	Guangdong6	Guangxi1	Guangxi2	Guangxi3	Guangxi4	Guangxi5	Guangxi6	Yunnan1	Yunnan2	Yunnan3	Yunnan4	Yunnan5	Yunnan6
1	2,3-butanediol	265.09	184.68	76.15	209.6	83.16	179.33	296.97	0	114.44	256.59	81.45	1019.88	173.87	256.31	466.38	410.5	165.52	50.48
2	propylene glycol	0	0	0	0	0	0	0	0	75.58	0	0	0	0	0	0	0	0	0
3	hexanal	399.76	279.98	313.08	308.33	150.05	260.23	230.54	57.51	130.21	294.62	209.07	182.99	207.86	36.23	231.15	241.62	155.67	182.12
4	(*E*)-2-nonenal	56.2	95.35	68.21	71	36.96	85.51	0	59.88	34.18	60.66	73.78	79.64	44.27	70.81	109.32	80.28	0	0
5	benzaldehyde	57.98	109.82	50.92	0	0	168.28	0	0	46.19	102.28	52.62	94.52	0	49.57	69.14	58.67	54.31	0
6	3,5-dimethoxy-4-hydroxybenzaldehyde	4115.06	3970.11	3258.76	4530.25	1861.44	1393.28	5558.03	0	0	0	0	0	4106.21	2356.17	4986.34	1882.87	1255.94	3220.19
7	2,6-di-tert-butyl-4-methylphenol	6272.71	6702.74	6540.27	5238.74	4705.8	5559.88	6468.03	7106.41	6733.2	5149.51	5177.12	7033.33	6293.4	5659.68	7638.6	7938.76	5115.72	3851.64
8	4-allyl-2,6-dimethoxyphenol	0	0	0	0	0	72.86	440.73	0	243.43	0	0	0	129.33	0	0	269.6	132.49	0
9	2-methoxy-4-acetylphenol	0	0	0	0	352.34	0	693.73	808.51	638.98	1094.03	935.46	584.08	855.15	0	0	0	669.33	649.86
10	acetic acid	10,184.79	10,757.27	2778.13	6350.81	6432.09	5900.64	6650.69	4210.8	2977.92	10,999.92	5047.46	11,141.62	4237.56	1068.38	12,748.83	9209.62	5570.25	10,334.03
11	methanoic acid	249.89	123.64	219.04	195.29	84.47	96.96	0	0	0	0	0	0	91.75	0	389.25	426.15	0	286.39
12	propanoic acid	821.24	619.01	644.68	396.79	303.37	260.37	332.73	147.19	329.19	457.22	280.06	653.58	425.26	299.24	1143.61	994.93	515.4	344.61
13	butanoic acid	2660.81	2644.03	614.21	2513.84	1227.91	1421.39	810.96	192.43	1269.77	1178.36	168.86	440.72	1266.55	837.56	4204.64	2118.54	1154.6	0
14	3-methylbutanoic acid	1633.63	1076.75	645.7	306.69	0	12.9	271.65	116.04	218.33	884.46	810.07	710.73	1123.49	566.92	1390.95	1609.2	0	0
15	2-methylbutanoic acid	0	0	0	0	0	373.25	0	0	0	0	0	0	0	0	0	0	0	0
16	2-methylpentanoic acid	0	0	0	183.26	0	37.89	0	0	0	0	0	0	0	0	0	0	302.4	0
17	4-methylpentanoic acid	0	0	0	0	0	0	0	0	0	0	0	0	0	0	0	0	119.8	56.28
18	hexanoic acid	300.51	597.73	487.13	280.13	283.56	435.09	280.31	0	155.31	453.98	119.62	112.67	580.84	441.59	727.74	728.69	383.25	78.31
19	octanoic acid	228.77	267.6	0	0	0	0	0	0	0	0	0	0	0	262.06	252.12	487.06	0	0
20	nonanoic acid	0	0	0	0	0	413.1	0	0	0	0	0	0	0	0	0	0	0	227.94
21	benzoic acid	3080.65	3564.46	4062.78	2709.59	1609.97	1795.59	3194.93	1982.81	2736.59	3573.53	1480.4	2837.5	2965.81	1857.28	4446.03	4751.21	1872.63	2262
22	phenylacetic acid	1468.96	1173.77	1187.97	880.89	730.3	47.37	1267.41	0	945.3	1180.95	741.94	1353.01	526.12	903.11	1487.08	1976.49	747.98	606.57
23	3-phenylpropionic acid	0	0	0	156.03	0	10.79	0	0	0	0	0	0	0	0	0	0	0	0
24	pentadecanoic acid	904.37	1205.19	867.02	423.32	0	806.02	586.5	0	0	0	0	0	0	733.11	1093.45	886.36	381.02	0
25	2-methyl-4,5-dihydro-3(2*H*)-furanone	341.22	1177.87	513.32	276.67	70.39	327.17	202.04	291.29	93.57	712.73	141.35	173.07	701.59	1141.42	391.82	578.35	726.04	1389.22
26	1-hydroxy-2-propanone	457.94	582.3	605.26	438.74	206.31	652.59	841.13	715.56	503.38	505.1	220.33	211.55	458.82	360.04	382.45	461.7	355.89	438.74
27	1-hydroxy-2-butanone	39.22	111.06	41.37	43.91	0	16.7	0	76.4	82.88	62.58	0	24.26	56.34	46.29	63.27	56.43	0	0
28	1-acetoxy-2-propanone	118.22	230.25	187.71	168.77	0	66.52	218.3	159.99	115.45	245.41	139.08	235.11	176.14	191.49	132.61	177.62	125.3	131.45
29	2(5*H*)-furanone	475.08	581.86	479.55	0	0	145.49	394.39	265.72	286.88	221.84	203.23	363.54	0	232.91	712.75	266.08	0	154.22
30	3-methyl-1,2-cyclopentanedione	0	0	0	0	0	0	0	0	0	0	0	0	0	0	0	0	0	361.67
31	2-hydroxy-3-methyl-2-cyclopenten-1-one	468.6	482.48	707.42	283.22	182.61	588.56	879.99	726.67	745.18	519.02	186.46	476.39	681.71	637.34	716.33	684.63	248.07	0
32	2(3*H*)-furanone	974.6	721.47	1069.93	351.84	292.08	470.52	631.95	615.99	655.84	705.84	496.09	977.5	560.98	596.89	1123.58	1324.65	397.11	323.38
33	2,5-dimethyl-4-hydroxy-3(2*H*)-furanone	271.76	326.77	454.38	230.03	123.31	731.6	2168.59	1158.64	298.58	471.8	69.64	144.05	415.29	861.94	625.51	529.53	0	141.69
34	4-hydroxy-5-methyl-3-(2*H*)-furanone	0	0	0	181.54	0	19.69	612.25	0	0	0	0	0	0	0	0	0	0	99.22
35	4-hydroxyacetophenone	1461.87	1892.62	1469.22	0	0	0	1546.24	0	0	0	0	0	0	1656.02	2234.53	1370.88	1132.23	0
36	2,5-dimethylpyrazine	924.49	1938.05	111.73	963.3	201.81	1827.76	1343.97	989.09	1309.65	1748.65	88.92	70.87	1114.25	703.42	419.79	430.64	375.15	93.56
37	2,6-dimethylpyrazine	682.76	1035.34	144.76	983.96	72.49	1094.56	1444.15	1045.87	800	946.87	68.92	85.72	693.95	273.74	230.77	289.35	139.77	208.04
38	2,3,5-trimethylpyrazine	0	0	0	180.76	35.78	306.68	249.04	250.84	40.86	0	46.57	0	0	125.18	0	0	0	33.86
39	2,5-dimethyl-3-ethylpyrazine	151.21	315.82	0	157.83	0	338.49	0	0	0	0	0	0	109.14	201.75	138.27	119	0	0
40	2,6-dimethyl-3-ethylpyrazine	0	0	0	0	79.78	186.16	412.18	272.34	203.53	286.15	0	0	0	0	0	0	82.88	0
41	2-acetyl-5-methylpyrazine	0	0	0	156.34	0	218.85	0	0	0	0	0	0	0	0	0	0	0	0
42	2-acetyl-6-methylpyrazine	251.42	419.45	0	141.33	0	245.73	183.55	144.23	121.93	218.58	0	0	0	408.61	0	113.67	96.94	0
43	2-acetylfuran	0	0	0	0	0	5.27	0	231.24	103.74	0	37.99	72.73	0	0	0	0	0	54.65
44	phenylethylene	651.29	613.81	573.68	451.99	160.07	457.61	546.32	469.44	511.72	984.77	984.42	1025.24	500.79	459.13	564.51	601.32	471.01	162.2
Cosine of included angle		0.9879	0.9888	0.8855	0.9800	0.9750	0.9643	0.9031	0.9463	0.9152	0.9527	0.9776	0.9671	0.9439	0.8155	0.9815	0.9839	0.9822	0.9189
Similarity		0.9850	0.9859	0.8562	0.9752	0.9781	0.9553	0.8824	0.9373	0.8980	0.9445	0.9762	0.9664	0.9300	0.7655	0.9773	0.9799	0.9787	0.9138

## Data Availability

Not applicable.

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
