# Peer review of "Analysis and Comparison of Aroma Compounds of Brown Sugar in Guangdong, Guangxi and Yunnan Using GC-O-MS"

_molecules, 2022, doi:10.3390/molecules27185878_

Round 1

Reviewer 1 Report

The manuscript titled “Analysis and Comparison of Aroma Compounds of Brown Sugar in Guangdong, Guangxi and Yunnan using GC-O-MS” deals with the differences between odor compounds of brown sugar from Guangdong, Guangxi, and Yunnan provinces in China. The subject is interesting and presents some novelty. However, there are some weak points in the research, concerning overall design of the experiment and analytical chemistry. The main criticism for my side is related to materials and method sections which are needed improvement. The analysis methods used are well documented, but do not include precise instrumental analysis. I recommend the manuscript for publication in its not present state in the Molecules, it needs some revisions.

My remarks about the text are as follows

In 2.1. Materials

-          Add the obtaining year of the eighteen brown sugar samples.

In 2.3. Extraction of odor compounds from sugars

-          The concentration of volatile compounds shown in Tables are by reference to a single internal standard (2-methy-3-heptanone). Is it enough only one internal standard for all compounds? What is the recovery yield of the internal standard?

In 2.4. GC-MO analysis

-          Add the model of sniffing port (ODP-2, 3, or 4)

-          - Identification took place by using only one DB-Wax ms capillary column. For proper verification of identification, it is necessary to compare retention indices on two columns of different polarities. In most cases, substances, which co-elute on one column can be distinguished on the other column with different polarities. Please explain

Author Response

The manuscript is revsied according to your comment, please have a check.

Reviewer 2 Report

The work is very well done.  
I have no comments to plan and execute GC analyzes.  The statistics explain the goals set, but are not readily legible.  The authors should add a multivariate PCA analysis of all sugar samples.  Then the statistics will be full.  From the PCA analysis, determine the differentiating factors and compare them with OPLS-DA. 

Author Response

The manuscript is revised according to your comments, please have a check.

Round 2

Reviewer 2 Report

Accept in present form

Author Response

Thanks for your reviews.